# The Role of Sonic Hedgehog Signaling in the Tumor Microenvironment of Oral Squamous Cell Carcinoma

**DOI:** 10.3390/ijms20225779

**Published:** 2019-11-17

**Authors:** Kiyofumi Takabatake, Tsuyoshi Shimo, Jun Murakami, Chang Anqi, Hotaka Kawai, Saori Yoshida, May Wathone Oo, Omori Haruka, Shintaro Sukegawa, Hidetsugu Tsujigiwa, Keisuke Nakano, Hitoshi Nagatsuka

**Affiliations:** 1Department of Oral Pathology and Medicine, Graduate School of Medicine, Dentistry and Pharmaceutical Sciences, Okayama University, Okayama 7008525, Japande18018@s.okayama-u.ac.jp (H.K.); de20052@s.okayama-u.ac.jp (S.Y.); p1qq7mbu@s.okayama-u.ac.jp (M.W.O.); p4628fuz@s.okayama-u.ac.jp (O.H.); gouwan19@gmail.com (S.S.); tsuji@dls.ous.ac.jp (H.T.); pir19btp@okayama-u.ac.jp (K.N.); jin@okayama-u.ac.jp (H.N.); 2Division of Reconstructive Surgery for Oral and Maxillofacial Region, Department of Human Biology and Pathophysiology, School of Dentistry, Health Sciences University of Hokkaido, Hokkaido 0610293, Japan; shimotsu@hoku-iryo-u.ac.jp; 3Department of Oral and Maxillofacial Radiology, Graduate School of Medicine, Dentistry and Pharmaceutical Sciences, Okayama University, Okayama 7008525, Japan; jun-m@md.okayama-u.ac.jp; 4Department of Anatomy, Basic Medical Science College, Harbin Medical University, Harbin 150081, China; 5Department of Oral and Maxillofacial Surgery, Kagawa Prefectural Central Hospital, Kagawa 7608557, Japan; 6Department of Life Science, Faculty of Science, Okayama University of Science, Okayama 7000005, Japan

**Keywords:** sonic hedgehog (SHH), oral squamous cell carcinoma (OSCC), tumor microenvironment (TME), tumor-associated macrophages (TAMs), cancer-associated fibroblasts (CAFs), tumor-associated angiogenesis

## Abstract

Sonic hedgehog (SHH) and its signaling have been identified in several human cancers, and increased levels of SHH expression appear to correlate with cancer progression. However, the role of SHH in the tumor microenvironment (TME) of oral squamous cell carcinoma (OSCC) is still unclear. No studies have compared the expression of SHH in different subtypes of OSCC and focused on the relationship between the tumor parenchyma and stroma. In this study, we analyzed SHH and expression of its receptor, Patched-1 (PTCH), in the TME of different subtypes of OSCC. Fifteen endophytic-type cases (ED type) and 15 exophytic-type cases (EX type) of OSCC were used. H&E staining, immunohistochemistry (IHC), double IHC, and double-fluorescent IHC were performed on these samples. ED-type parenchyma more strongly expressed both SHH and PTCH than EX-type parenchyma. In OSCC stroma, CD31-positive cancer blood vessels, CD68- and CD11b-positive macrophages, and α-smooth muscle actin-positive cancer-associated fibroblasts partially expressed PTCH. On the other hand, in EX-type stroma, almost no double-positive cells were observed. These results suggest that autocrine effects of SHH induce cancer invasion, and paracrine effects of SHH govern parenchyma-stromal interactions of OSCC. The role of the SHH pathway is to promote growth and invasion.

## 1. Introduction

Oral squamous cell carcinoma (OSCC) is a malignant tumor that comprises up to 90% of tumors in the head and neck region [1] and is a heterogeneous group of tumors arising from the mucosal surfaces of the oral cavity.

OSCC is classified into various subtypes as described in the World Health Organization (WHO) Classification of Head and Neck Tumors 4th Ed [2]. In addition, macroscopic subtypes have also been identified, based on the clinical invasion pattern. These subtypes have important differences in prognosis due to differences in invasive ability. Clinically, tumors primarily show exophytic growth (exophytic type—EX type) or endophytic growth (endophytic type—ED type).

OSCC is a malignant epithelial tumor. Similar to many other solid tumors, the tumor microenvironment (TME) of OSCC consists of the tumor parenchyma and stroma. The stroma of the TME is composed of multiple different cell types, such as macrophages, endothelial cells, cancer-associated fibroblasts (CAFs), and immune cells. These subpopulations of cells interact with each other as well as with cancer cells via complex communication networks through various secreted cytokines, chemokines, growth factors, and proteins of the extracellular matrix.

Macrophages, especially tumor-associated macrophages (TAMs), largely contribute to proliferation, invasion, and metastasis of the tumor. Recently, some studies have suggested that a relationship exists between the level of infiltration of TAMs and a poor outcome of OSCC and that the relationship could be used as a potential prognostic marker [3,4,5]. CD11b is monocytes macrophage lineage-specific marker. Numerous studies have reported that CD11b plays a role in invasion and metastasis and CD11b is a marker of TAMs in tumors [6].

Blood vessels in the tumor stroma play an important role. Tumor blood vessels nourish not only cancer cells but also stromal cells. Tumor blood vessels are positive for CD31, which was first characterized as a protein that is expressed by human hematopoietic progenitor cells and has been considered a definitive marker of angiogenesis in neoplastic lesions [7]. Recently, it has been reported that CXCR4 plays a crucial role in tumor angiogenesis, which is required for OSCC progression [8]. Moreover, stromal cell-derived factor 1 (SDF-1), which is the ligand for CXCR4, is also expressed in various types of cancers [9,10].

Additionally, CAFs are the predominant cell type within the tumor stroma, and their main function is to maintain a favorable microenvironment for tumor cell growth and proliferation. The most common marker used to detect CAFs in the tumor stroma is α-smooth muscle actin (α-SMA), a specific marker of myofibroblasts [11,12,13,14]. This myofibroblast phenotype of CAFs is frequently observed in OSCC, and upregulation of α-SMA is correlated with poor prognosis [15].

Thus, the biological properties of the tumor stroma are closely related to the growth and spread of cancer. However, the details of how the TME varies depending on the proliferation type (such as ED or EX type) of OSCC have not been reported.

Hedgehog signaling is essential for proper pattern formation and morphogenesis during embryogenesis [16,17,18,19]. The interaction of the Hedgehog protein with its receptor, Patched-1 (PTCH), leads to activation of the transcription factor, Gli, which induces downstream target genes including PTCH and Gli themselves [20]. Among the three types of Hedgehog genes, sonic hedgehog (SHH), Indian hedgehog, and desert hedgehog, the role of the SHH pathway in blood vessel formation has been well studied in several animal models [21,22]. SDF-1 recruits endothelial progenitors, and SHH increases the expression of SDF-1 in a concentration-dependent manner [23].

Hedgehog signaling contributes to the development and progression of many cancers [24,25,26,27,28,29,30,31,32]. In OSCC, reactivation and overexpression of the Hedgehog pathway plays a key role in development and progression [33,34,35,36], and SHH signal activation is associated with worse prognosis in OSCC [37,38].

However, how Hedgehog signaling is involved in the TME of OSCC is unclear, and no studies have focused on the expression of SHH signaling according to the subtype (such as ED or EX type) of OSCC.

In the present study, we analyzed the relationship between SHH signaling and TEM in OSCC comparing resected ED-type or EX-type OSCC samples.

## 2. Results

### 2.1. SHH and PTCH Expression in OSCC

To investigate how the expression of SHH and its receptor PTCH influence on the invasion of OSCC, we examined the expression of SHH and PTCH in the parenchymal and stromal regions comparing ED-type and EX-type specimens.

As seen with H&E staining, the existing basal membrane in ED type was destroyed and infiltrated by forming cancer nests in the subepithelial connective tissue (Figure 1a). On the other hand, the cancer cells did not invade over the existing basal membrane, and cancer cells growth was observed as extrovert in the EX type (Figure 1b).

In the cancer parenchyma, SHH was strongly expressed in the cytoplasm of cancer cells in ED type: Moderate (+2), five patients; marked (+3), 10 patients, and more strongly in the cancer nests at the front line of invasion (Figure 1c). On the other hand, in EX type, strong expression was observed in the cytoplasm of part of the basal layer, and expression was observed from the spinous layer to the surface layer. However, the expression was weaker than that in ED type: Weak (+1), four patients; moderate (+2), 11 patients (Figure 1d).

In the cancer parenchyma, PTCH was strongly expressed in the cytoplasm of cancer cells in ED type: Moderate (+2), five patients; marked (+3), 10 patients, particularly in the cancer nests at the invasion front (Figure 1e). On the other hand, the cytoplasm of cancer cells from the basal layer to the surface layer was weakly positive in EX type: No reactivity (0), one patient; weak (+1), 12 patients; moderate (+2), two patients, (Figure 1f).

SHH in the cancer stroma was strongly expressed in blood vessels, spindle cells, and round cells in ED type (Figure 1g). SHH in EX-type stroma was also strongly expressed in blood vessels and round cells (Figure 1h). The expression of PTCH in the cancer stroma was stronger in EX type than in ED type, and blood vessels, round cells, and spindle-shaped cells were positive for PTCH (Figure 1i,j).

### 2.2. Involvement of SHH Signaling in Tumor Progression

#### 2.2.1. Tumor Angiogenesis

To clarify whether the SHH pathway is involved in angiogenesis in OSCC, we first determined the effect of SHH/PTCH in neovascularization using CD31 IHC staining. CD31 is a marker associated with angiogenesis and labels vascular endothelial cells of blood vessels in tumors. Vascular endothelial cells of blood vessels were positive for CD31 in both ED-type and EX-type cancer stroma. Especially in ED-type stroma, CD31-positive expression was observed not only in blood vessels but also round cells (Figure 2a). When comparing the area of blood vessels in the cancer stroma between ED type and EX type, ED type showed superior angiogenesis compared to EX type (Figure 2b).

Many tumor blood vessels, which expressed both SHH and PTCH, were observed in the cancer stroma, and the accumulation of these blood vessels was adjacent to the site where SHH was strongly expressed in the cancer parenchyma (Figure 2c). However, connective tissue adjacent to epithelial tissue in the non-cancerous region had few blood vessels that expressed SHH (Figure 2d). The expression of PTCH in tumor blood vessels also showed the same tendency as the expression of SHH, and PTCH was not expressed in blood vessels in the connective tissue adjacent to the non-cancerous region epithelium (Figure 2e,f).

In double-fluorescent IHC staining, both PTCH and CD31 were positive in blood vessels in the cancer stroma in ED type and EX type. In addition, accumulation of CD31-positive round cells was observed around blood vessels in the stroma of ED type, however these cells did not express PTCH (Figure 2g). EX type also showed PTCH expression in almost all CD31-positive blood vessels in the cancer stroma (Figure 2h).

SDF-1/CXCR4 plays multiple roles in tumor pathogenesis. CXCR4 promotes tumor growth and malignancy, enhances tumor angiogenesis, and participates in tumor metastasis [39,40,41,42]. SDF-1 was expressed in both ED-type and EX-type cancer parenchyma, and the expression was stronger in ED type than in EX type. SDF-1 expression was observed in the cancer parenchyma; however, SDF-1 was not expressed in the non-cancerous epithelium adjacent to the cancer parenchyma, similar to SHH expression (Figure 3a). Importantly, more intense signals of SDF-1 were detected in the microvascular cells at the cancer’s invasive front (Figure 3b). In EX type, SDF-1 expression was detected in blood vessels of stroma (Figure 3c). CXCR4 was also expressed in both ED-type and EX-type cancer parenchyma; however, CXCR4 was not expressed in the non-cancerous epithelium adjacent to the cancer parenchyma, similar to PTCH expression (Figure 3d). In ED type, CXCR4 were detected in the blood vessels at the cancer stroma (Figure 3e); however, in EX type, CXCR4 expression was not detected in blood vessels of stroma (Figure 3f). Double-fluorescent IHC staining showed that these CXCR4-positive structures expressed PTCH in ED type (Figure 3g).

#### 2.2.2. Tumor-Associated Macrophages

To clarify whether the SHH pathway is involved in OSCC invasion via TAMs, we first determined the effect of SHH/PTCH in OSCC invasion using CD68 IHC staining. CD68 is a pan-macrophage marker and is considered a marker of TAMs. Recently, CD68-positive cells have been reported to be a poor prognostic factor [5] and TAMs recruitment in SHH expression tumor recruit TAMs significantly [43].

In both ED and EX type, accumulation of CD68-positive round or dendritic-shaped cells was observed in the cancer stroma (Figure 4a). The number of CD68-positive cells in the cancer stroma was significantly higher in ED type than in EX type (Figure 4b). Double immunostaining for SHH and CD68 showed that the expression of these molecules did not merge. However, a large concentration of CD68 was observed at the site close to the cancer parenchyma where SHH expression was strong (Figure 4c). Double-fluorescent IHC staining for CD68 and PTCH showed abundant CD68-positive cells that overlapped with PTCH in ED type (Figure 4d), whereas almost all CD68-positive cells did not merge with PTCH in EX type.

CD11b is a specific monocyte macrophage marker, and recently it was reported that CD11b is a marker of macrophages, some of which are TAMs in tumors [44]. Thus, we next determined the effect of SHH/PTCH in OSCC invasion using CD11b IHC staining. In both ED type and EX type, aggregation of CD11b-positive cells was observed in the cancer stroma. Many round CD11b-positive cells were observed in the ED-type cancer stroma, and CD11b-positive round cells were also observed in the EX-type cancer stroma (Figure 4e). The number of CD11b-positive cells was higher in ED type than EX type (Figure 4f). Double immunostaining for SHH and CD11b showed that the expression of these molecules did not merge. However, a large concentration of CD11b was present at the site adjacent to the cancer parenchyma where SHH expression was strong (Figure 4g). When the expression of PTCH and CD11b was examined, CD11b-positive round cells merged with PTCH expression (Figure 4h). On the other hand, almost no expression of PTCH merged with CD11b-positive round cells in the EX-type stroma.

#### 2.2.3. Cancer-Associated Fibroblasts

To clarify whether the SHH pathway is involved in OSCC invasion via CAFs, we investigated the effect of SHH/PTCH in OSCC invasion using α-SMA IHC staining. α-SMA is a common marker of myoepithelial cells, especially CAFs, in many human tumors. In ED-type stroma, abundant spindle-shaped cells were observed around the cancer nests, and these cells were positive for α-SMA (Figure 5a). In EX-type stroma, α-SMA-positive blood vessels were observed in the cancer stroma. However, almost no spindle-shaped α-SMA-positive cells were observed (Figure 5b). Double immunostaining for SHH and α-SMA showed that the expression of these molecules did not overlap. However, a large concentration of α-SMA was seen at the site close to the cancer parenchyma where SHH expression was strong (Figure 5c).

Double-fluorescent IHC staining showed that some α-SMA-positive spindle-shaped cells expressed PTCH in ED-type stroma (Figure 5d). PTCH-positive cells were observed in EX-type cancer stroma, but no positive cells that overlapped with α-SMA were observed.

## 3. Discussion

In this study, the role of SHH in OSCC TME was clarified by comparing the expression of SHH by ED type and EX type by immunostaining.

First, we demonstrated autocrine expression of SHH in OSCC cancer parenchyma. Our data showed that both SHH and PTCH were expressed in the cancer parenchyma, thus suggesting that SHH signaling may operate in an autocrine manner in the cancer parenchyma. The expression of SHH and PTCH was higher in ED-type cancer parenchyma than in EX-type parenchyma, suggesting that SHH signaling may be involved in not only cancer growth but also cancer invasion. Autocrine SHH signaling depends on secretion of SHH ligands by the cancer parenchyma. SHH acts on itself in a positive feedback loop. Active Hedgehog signaling has been reported in various cancers such as prostate, lung, liver, thyroid, bladder, ovarian, and colon cancers [45,46,47,48,49,50,51]. SHH overexpression promotes tumor growth and metastasis, and higher levels of SHH are associated with poor survival and poor prognosis. Some studies reported that autocrine activation of SHH is present in colorectal cancer and breast cancer [52,53,54,55]. In addition, the SHH pathway induces epithelial-to-mesenchymal transition in gastric tumors, pancreatic cancer, and breast cancer [56,57,58]. Our data also showed that SHH expression was different in ED-type parenchyma compared to EX-type parenchyma. Therefore, SHH signaling in the cancer parenchyma of OSCC may be involved in causing epithelial-to-mesenchymal transition and invasion, similar to other tumors.

Second, we demonstrated paracrine signaling in which the cancer parenchyma secretes SHH ligands that bind receptors on the surrounding stroma, thus activating stromal SHH signaling.

The role of SHH during tumor-associated angiogenesis has not been clarified in OSCC. Here, we demonstrated a novel role for SHH in the development of the tumor vasculature. Our results imply that SHH secreted from cancer cells facilitates cancer invasion not only by stimulating proliferation of cancer cells in an autocrine manner but also by promoting angiogenesis in a paracrine manner. Angiogenesis of tumors has critical effects on development of the tumor. Some studies have reported the details of the contribution of SHH to tumor angiogenesis [59] and also reported the contribution of SHH to angiogenesis of OSCC [33,34]. PTCH expression was not expressed in blood vessels observed in the connective tissue adjacent to the normal epithelium. However, PTCH-positive tumor blood vessels were observed abundantly in sites close to the cancer parenchyma where SHH was strongly expressed. Thus, these results suggest that SHH acts in a paracrine manner on tumor blood vessels in a concentration-dependent manner. Additionally, our studies also showed that SHH was expressed in tumor blood vessels. Thus, SHH signaling had both autocrine and paracrine effects on tumor angiogenesis in OSCC.

Both CXCR4 and SDF-1 were expressed in the cancer cells themselves. SDF-1 expression is related to ovarian tumorigenesis and malignant transformation [60]. Pancreatic cancer [61], neuroblastoma cells [62], and glioblastoma [63] also express both CXCR4 and SDF-1 proteins. In addition, SDF-1 recruits endothelial progenitors, and SHH increases the expression of SDF-1 in a concentration-dependent manner [23]. Recently, some studies have indicated that CXCR4 is essential for the formation of large blood vessels that feed the gastrointestinal tract in the fetal stage [64,65]. PTCH-positive blood vessels have the potential to be a therapeutic target because CXCR4 is expressed in specific tumor blood vessels [8].

The SHH signaling pathway is not only involved in regulation of tumor angiogenesis but is also important for tumor migration and invasion. The number of macrophages that infiltrated the cancer stroma and expressed CD68 was significantly higher in ED-type stroma than in EX-type stroma. In other tumors, the high accumulation of CD68-positive cells may be associated with poor prognosis [3,66], suggesting that CD68 may also be correlated with prognosis of OSCC. In addition, CD68 overlapped with PTCH expression in ED-type stroma, suggesting that SHH signaling may be involved in macrophage aggregation in invasive cancer. Shimo et al. reported that the progression of oral cancer to the bone marrow of the jaw is correlated with prognosis, and that SHH produced from cancer cells directly affects CD68-positive osteoclast precursor cells and mature osteoclasts [67].

CD11b is generally known as a marker of monocytes, macrophages, and TAMs [5]. TAMs are involved in tumor invasion and metastasis [68]. In our study, CD11b-positive cells that contacted the cancer parenchyma were observed in ED-type stroma and EX-type stroma, and the number of CD11b-positive cells was higher in the former than the latter type. In ED type, some CD11b-positive cells were PTCH positive. On the other hand, in EX type, almost no CD11b-positive cells were PTCH positive. ED type that expressed SHH strongly induced CD11b/PTCH-double positive cells, which participate in cancer invasion around the front of cancer nests. Considering the characteristics of their distribution and shape, CD11b-positive cells in OSCC may represent TAMs, and some CD11b-positive cells may have been recruited by SHH signaling in the cancer parenchyma.

α-SMA is a popular marker of myoepithelial cells and CAFs in tumors. In our study, we found many spindle-shaped α-SMA-positive cells surrounding the cancer parenchyma in ED-type stroma. In ED-type stroma, some of these spindle-shaped α-SMA-positive cells expressed PTCH. On the other hand, we did not observe α-SMA-positive cells surrounding the cancer parenchyma in EX-type stroma. Thus, the SHH pathway has been identified as activated in CAFs in OSCC, especially in ED type. Previous work identified SHH as a mediator of the desmoplastic response in pancreatic cancer and suggested that the stroma may serve as a barrier to delivery of therapeutic compounds [69]. In a previous in vivo study, human pancreatic CAFs overexpressed the Hedgehog receptor Smoothened (SMO). Increased SMO expression indicates increased Hedgehog pathway activity in these cells, suggesting evidence for Hedgehog pathway activity in pancreatic cancer–associated stromal cells [70]. Our study is consistent with these previous reports and indicates that the SHH pathway in OSCC may play an important role in inducing CAFs through a paracrine mechanism.

The cancer stroma consists of various types of cells. Some cancer stromal cells are derived from BMDCs [71,72]. These stromal cells derived from BMDCs are recruited by SDF-1 and induced by SHH. Stromal cells differentiate from endothelial cells, TAMs, and CAFs. Therefore, SHH signaling affects not only tumor growth but also the cells that make up the cancer stroma and is an important pathway in the TME.

The limitations of this study are due to using the patients tissue and the lack of understanding of the cell biology and genetics approaches. Future research should include the cell biology and genetics approaches.

In conclusion, given all the findings we observed from OSCC of ED type and EX type, the SHH pathway may participate in the processes of tumorigenesis and cancer development. The histological comparison between ED type and EX type provides authentic evidence for the involvement of the SHH pathway in the development of cancer via differentiation into various kinds of cells in the cancer stroma, such as macrophages, fibroblasts, and endothelial cells. Our results suggest roles for these cells in tumorigenesis due to their multilineage differentiation potential. This study is, to the best our knowledge, the first to show the direct and indirect influence of the SHH pathway on the TME. Our findings clearly show that OSCC-derived SHH is involved in progression and invasion in the OSCC TME.

## 4. Materials and Methods

### 4.1. Patients and Samples

Patient samples were obtained from the Oral Pathology Department of Okayama University. This study was approved by the Ethics Committee of Okayama University Graduate School of Medicine, Dentistry and Pharmaceutical Sciences (the project identification code: 1608–018; date of approval: 10 March 2017; and name of the ethics committee: Analysis of biological property of oral cancer). A total of 30 cases of tongue OSCC classified as T2 according to the Union for International Cancer Control (UICC) 8th Ed criteria were enrolled in the retrospective study, including 15 ED-type cases and 15 EX-type cases of OSCC. These tissue samples from 30 patients were collected during excision. None of the patients received chemotherapy, radiotherapy, or immunotherapy before sampling.

Tissues were processed and embedded in paraffin wax according to routine histological preparation methods and sectioned at 3 μm thickness. The sections were used for hematoxylin-eosin (H&E) staining (Wako Pure Chemical Industries, Ltd., Osaka, Japan), immunohistochemistry (IHC), double IHC, and double-fluorescent IHC.

### 4.2. Immunohistochemistry and Evaluation

IHC was carried out using the antibodies detailed in Table 1. Following antigen retrieval, sections were treated with 10% normal serum for 30 min, and then incubated with primary antibodies at 4 °C overnight. Tagging of primary antibody was achieved by the subsequent application of anti-rabbit, anti-goat, or anti-mouse IgG and avidin–biotin complexes (Rabbit/Goat/Mouse ABC kit; Vector Laboratories, Inc., Burlingame, CA, USA). Immunoreactivity was visualized using diaminobenzidine (DAB)/H_2_O_2_ solution (Histofine DAB substrate; Nichirei, Tokyo, Japan), and sections were counterstained with Mayer’s hematoxylin. For double immunostaining, the same process from DAB staining was repeated, and immunoreactivity was visualized using Green chromogen (Vina Green Chromogen kit; Biocare Medical, Pacheco, CA, USA). Sections were counterstained with Mayer’s hematoxylin.

SHH and PTCH expression in OSCC were evaluated independently by the 2 pathologists. The results were scored from 0 to 3 based on the intensity of the staining at the membrane or in the cytoplasm: 0, no reactivity; +1, weak; +2, moderate; and 3, marked.

### 4.3. Double-Fluorescent IHC Staining

Double-fluorescent IHC for PTCH-CD31, PTCH-CD11b, PTCH-CXCR4, and PTCH-α-SMA was performed using PTCH monoclonal antibodies (goat IgG) (Abcam, Tokyo, Japan). The secondary antibodies applied are detailed in Table 2. Antibodies were diluted with Can Get Signal A (TOYOBO, Osaka, Japan). After antigen retrieval, sections were treated with Block Ace (DS Pharma Bio-medical, Osaka, Japan) for 20 min at room temperature. Specimens were incubated with primary antibodies at 4 °C overnight and then incubated with secondary antibodies (1:200) for 1 h at room temperature. After the reaction, the specimens were stained with 1 mg/mL DAPI (Dojindo Laboratories, Kumamoto, Japan).

### 4.4. Cell Counting

Sections were examined under a microscope at 400× magnification. Ten areas were randomly chosen in each sample. The number of positively labeled cells was counted manually, the average was obtained, and the ED type and EX type were compared.

### 4.5. Statistical Analysis

All values are the mean ± standard deviation. Statistical analysis was performed using one-way analysis of variance and Tukey’s tests. A *p* value <0.05 was considered significant. All calculations were made using PASW Statistics 18 (SPSS Inc., Chicago, IL, USA).

## Figures and Tables

**Figure 1 ijms-20-05779-f001:**
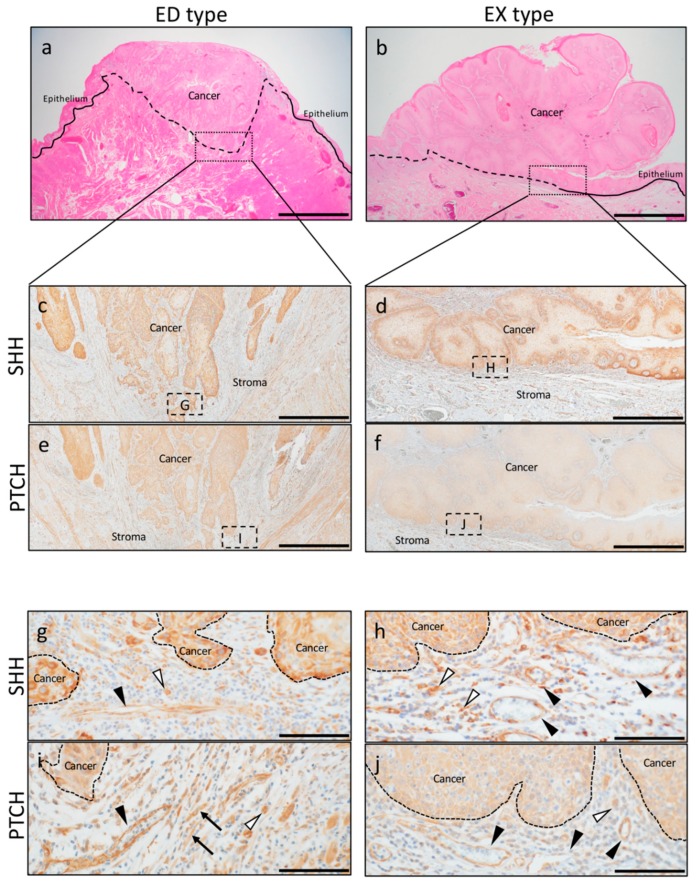
Representative pictures of immunohistochemical staining of sonic hedgehog (SHH) and Patched-1 (PTCH) in endophytic (ED)-type and exophytic (EX)-type oral squamous cell carcinoma (OSCC). (**a**,**b**) H&E staining of ED type and EX type. Scale bars: 2 mm. (**c**,**d**) Low-power magnification of SHH immunohistochemistry (IHC) staining. Scar bar: 1 mm. (**e**,**f**) Low-power magnification of PTCH IHC staining. Scale bar: 1 mm. (**g**,**h**) High-power magnification of SHH IHC staining. The expression of SHH in the cancer stroma was expressed in blood vessels (block arrowheads) and round cells (white arrowheads). Scale bar: 100 μm. (**i**,**j**) High-power magnification of PTCH IHC staining. The expression of PTCH in the cancer stroma was stronger in EX type than in ED type and was positive for blood vessels (block arrowheads), round cells (white arrowheads), and spindle-shaped cells (arrows). Scale bar: 100 μm.

**Figure 2 ijms-20-05779-f002:**
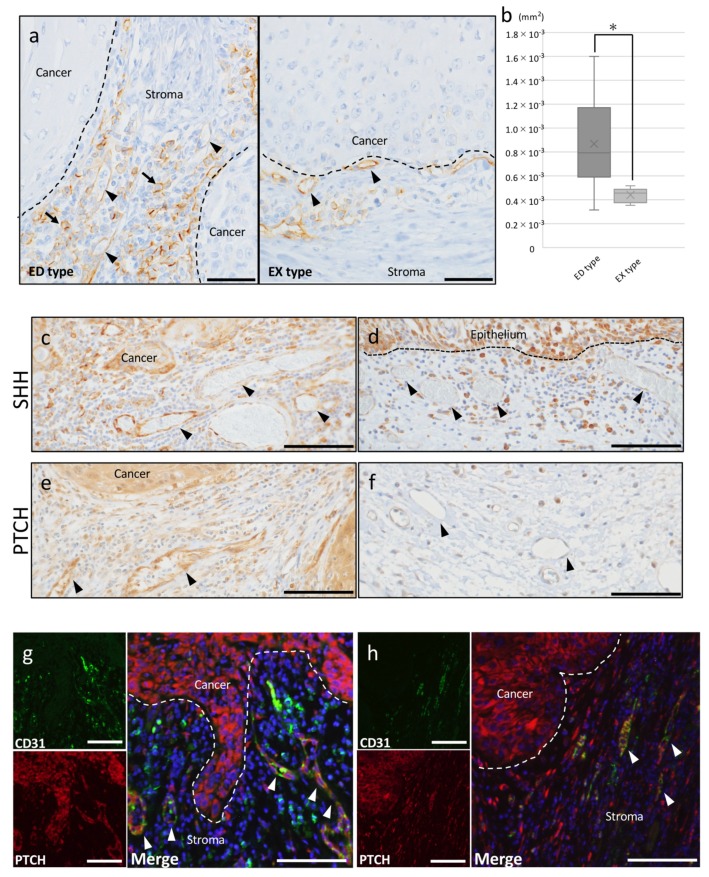
IHC and immunofluorescence of CD31. (**a**) IHC staining for CD31 in ED type and EX type. CD31 positive expression was observed in blood vessels (arrowheads). CD31 positive round-shape cells were observed (arrows). Scale bar: 50 μm. (**b**) Quantification of the angiogenesis area in ED type and EX type. ED type showed superior angiogenesis compared to EX type. * *p* < 0.05 as indicated. (**c**) IHC feature of SHH in ED-type stroma. The accumulation of these blood vessels (arrowheads) was shown adjacent to the site where the SHH was strongly expressed in the cancer parenchyma. Scale bar: 100 μm. (**d**) IHC feature of SHH in non-cancerous area. The connective tissue adjacent to normal epithelial tissue had few blood vessels that expressed SHH weekly (arrowheads). Scale bar: 100 μm. (**e**) IHC feature of PTCH in ED-type stroma. The PTCH positive blood vessels (arrowheads) was shown adjacent to the cancer parenchyma. Scale bar: 100 μm. (**f**) IHC feature of PTCH in non-cancerous area. The PTCH positive blood vessels were not observed (arrowheads). Scale bar: 100 μm. (**g**,**h**) Double-fluorescent IHC in ED type and EX type. Double-fluorescent IHC for PTCH-CD31 demonstrated that both PTCH and CD31 were positive in blood vessels (arrowheads) in the cancer stroma in ED type and EX type. Scale bar: 100 μm.

**Figure 3 ijms-20-05779-f003:**
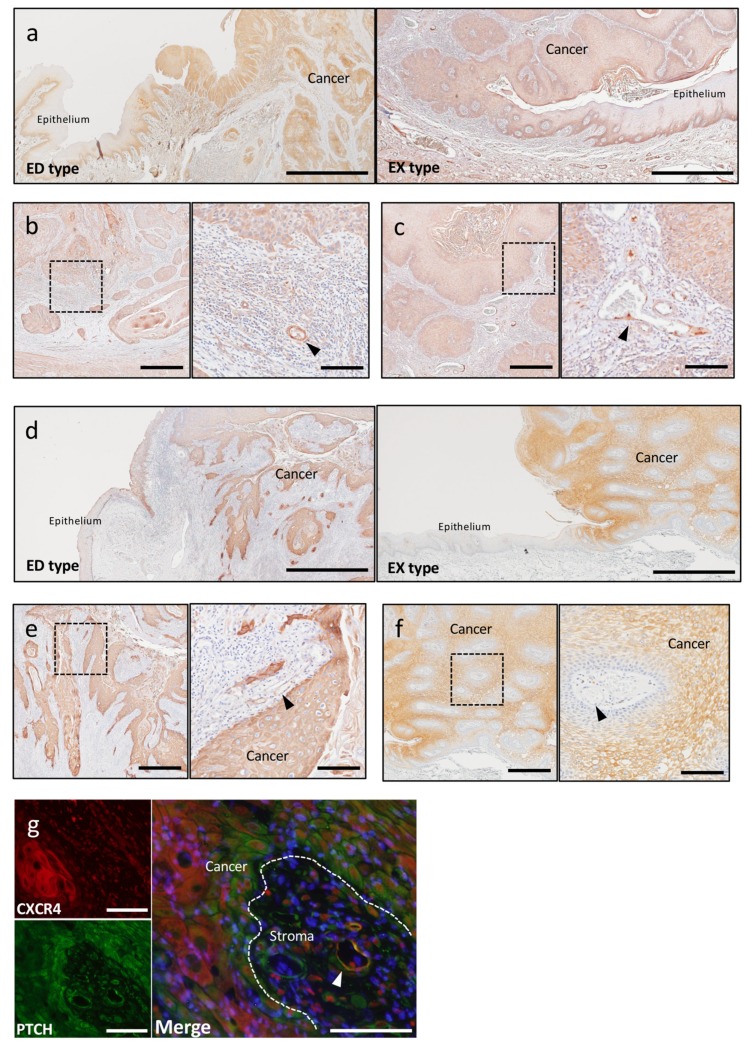
IHC and immunofluorescence of SDF-1 and CXCR4. (**a**) IHC staining for SDF-1 in ED type and EX type. SDF-1 was expressed in both ED-type and EX-type cancer parenchyma; however, SDF-1 was not expressed in the non-cancerous epithelium adjacent to the cancer parenchyma. Scale bar: 1 mm. (**b**) IHC feature of SDF-1 in ED type. Blood vessels (arrowhead) were positive for SDF-1 in invasive front of cancer. Scale bar, left: 500 μm, right: 100 μm. (**c**) IHC feature of SDF-1 in EX type. Blood vessels (arrowhead) were positive for SDF-1 weakly in cancer stroma. Scale bar, left: 500 μm, right: 100 μm. (**d**) IHC staining for CXCR4 in ED type and EX type. CXCR4 was expressed in both ED-type and EX-type cancer parenchyma; however, CXCR4 was not expressed in the non-cancerous epithelium adjacent to the cancer parenchyma. Scale bar: 1 mm. (**e**) IHC feature of CXCR4 in ED type. Blood vessels (arrowhead) were positive for CXCR4 in cancer stroma. Scale bar, left: 500 μm, right: 100 μm. (**f**) IHC feature of CXCR4 in EX type. Blood vessels (arrowhead) were not positive for CXCR4 in cancer stroma. Scale bar, left: 500 μm, right: 100 μm. (**g**) Double-fluorescent IHC in ED type. Double-fluorescent IHC staining of CXCR4 and PTCH demonstrated that the CXCR4-positive blood vessels merged with PTCH in ED type (arrowhead). Scale bar: 100 μm.

**Figure 4 ijms-20-05779-f004:**
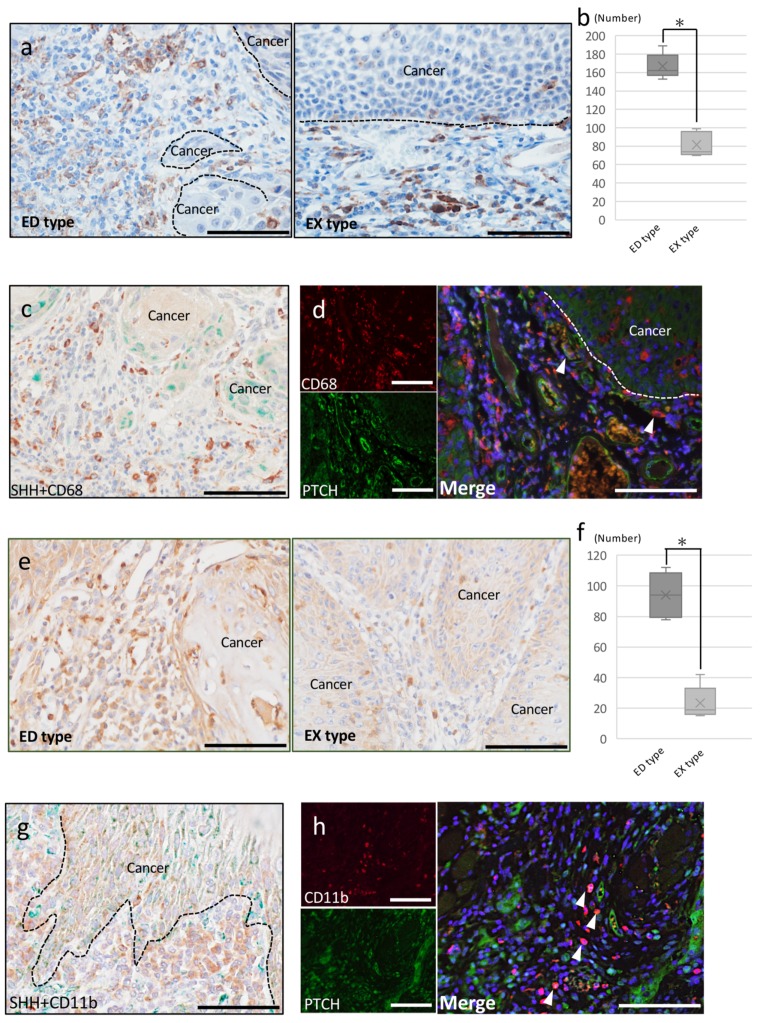
IHC, double IHC, and immunofluorescence of CD68 and CD11b. (**a**) IHC staining for CD68 in ED type and EX type. The accumulation of CD68-positive round- or dendritic-shape cells was observed in the cancer stroma. Scale bar: 100 μm. (**b**) Quantification of the number of CD68-positive cells in ED type and EX type. The number of CD68-positive cells in the cancer stroma was significantly higher in ED type than in EX type. * *p* < 0.05 as indicated. (**c**) Double IHC staining for SHH (green)-CD68 (brown) in ED type. The expression of SHH and CD68 did not merge. However, there was a large concentration of CD68 at the site close to the cancer parenchyma where SHH expression was strong. Scale bar: 100 μm. (**d**) Double-fluorescent IHC in ED type. Double-fluorescent IHC staining of CD68 and PTCH demonstrated that the abundant CD68-positive cells merged (arrowheads) with PTCH in ED type. Scale bar: 100 μm. (**e**) IHC staining for CD11b in ED type and EX type. The accumulation of CD11b-positive round- or dendritic-shape cells was observed in the cancer stroma. Scale bar: 100 μm. (**f**) Quantification of the number of CD11b-positive cells in ED type and EX type. The number of CD11b-positive cells in the cancer stroma was significantly higher in ED type than in EX type. * *p* < 0.05 as indicated. (**g**) Double IHC staining for SHH (green)-CD11b (brown) in ED type The expression of SHH and CD11b did not merge. However, there was a large concentration of CD68 at the site close to the cancer parenchyma where SHH expression was strong. Scale bar: 100 μm. (**h**) Double-fluorescent IHC in ED type. Double-fluorescent IHC staining of CD11b and PTCH demonstrated that the abundant CD11b-positive cells merged (arrowheads) with PTCH in ED type. Scale bar: 100 μm.

**Figure 5 ijms-20-05779-f005:**
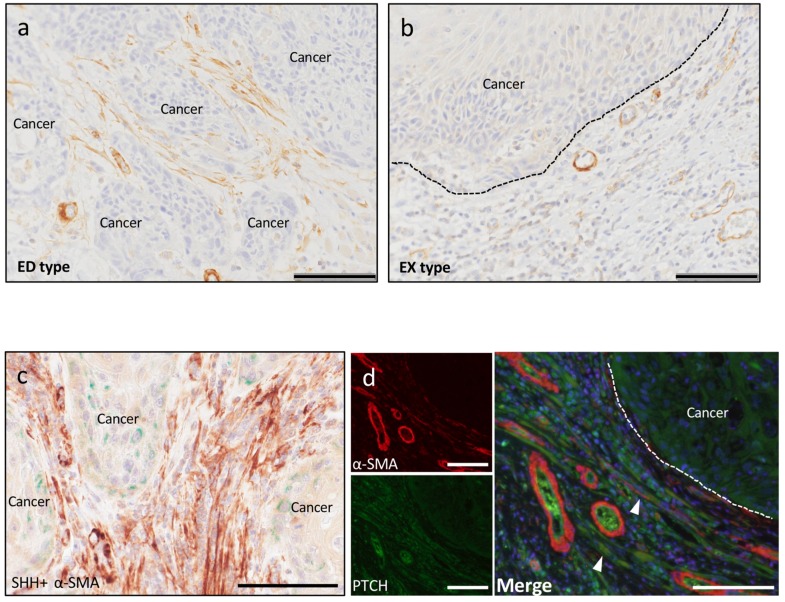
IHC, double IHC and immunofluorescence of α-smooth muscle actin (α-SMA). (**a**) IHC staining for α-SMA in ED type. The abundant α-SMA-positive spindle-shaped cells were observed around the cancer nests. Scale bar: 100 μm. (**b**) IHC staining for α-SMA in EX type. Almost all spindle-shaped α-SMA-positive cells were not observed. Scale bar: 100 μm. (**c**) Double IHC staining for α-SMA (green)-CD11b (brown) in ED type. There was a large concentration of α-SMA at the site close to the cancer parenchyma where SHH expression was strong. Scale bar: 100 μm. (**d**) Double-fluorescent IHC in ED type. Double-fluorescent IHC staining of α-SMA and PTCH demonstrated that the abundant α-SMA-positive cells merged (arrowheads) with PTCH partially in ED type. Scale bar: 100 μm.

**Table 1 ijms-20-05779-t001:** Antibodies used in immunohistochemistry.

Primary Antibody	Immunized Animal	Antigen Retrieval	Dilution	Supplier
SHH	Rabbit	Heated in 0.01 mol/L citrate buffer for 3 min	×100	Abcam (Tokyo, Japan)
PTCH	Goat	Heated in 0.01 mol/L citrate buffer for 3 min	×100	Abcam (Tokyo, Japan)
CD31	Mouse	Pressurized with 0.01 mol/L citrate buffer for 8 min in microwave oven	×100	Novocastra (Newcastle upon Tyne, UK)
SDF-1	Rabbit	Heated in 0.01 mol/L citrate buffer for 3 min	×200	Abcam (Tokyo, Japan)
CXCR4	Rabbit	Pressurized with 0.01 mol/L citrate buffer for 8 min in microwave oven	×300	Abcam (Tokyo, Japan)
CD68	Rabbit	Heated in 0.01 mol/L citrate buffer for 3 min	×200	Santa Cruz Biotechnology (Dallas, USA)
CD11b	Rabbit	Pressurized with 0.01 mol/L citrate buffer for 8 min in microwave oven	×500	Abcam (Tokyo, Japan)
α-SMA	Rabbit	Heated in 0.01 mol/L citrate buffer for 3 min	×200	Abcam (Tokyo, Japan)

**Table 2 ijms-20-05779-t002:** Antibodies used in double-fluorescent immunohistochemistry.

Second Antibody	Immunized Animal	Fluorescent Dye	Supplier
Anti-Rabbit IgG	Donkey	Alexa Flour 568	Thermo Fisher (Tokyo, Japan)
Anti-Goat IgG	Donkey	Alexa Flour 488	Thermo Fisher (Tokyo, Japan)
	Donkey	Alexa Flour 568	Thermo Fisher (Tokyo, Japan)
Anti-Mouse IgG	Donkey	Alexa Flour 568	Thermo Fisher (Tokyo, Japan)

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
