# Peer review of "The Role of Sonic Hedgehog Signaling in the Tumor Microenvironment of Oral Squamous Cell Carcinoma"

_ijms, 2019, doi:10.3390/ijms20225779_

Round 1

Reviewer 1 Report

Takabatake et al in the manuscript titled “The Role of Sonic Hedgehog Signaling in the Tumor  Microenvironment of Oral Squamous Cell Carcinoma“  performed H&E staining, immunohistochemistry (IHC), double IHC, and double-fluorescent IHC on samples of different  oral squamous cell carcinoma (OSCC) subtypes to study Sonic hedgehog (SHH),  expression of its receptor, Patched-1 (PTCH) in OSCC tumor microenvironment (TME). By using 15 endophytic-type cases (ED type) and 15 exophytic-type cases (EX type) cases they suggest that SHH –autocrine signaling induces growth promotion and cancer invasion. They also suggest that SHH paracrine effects are responsible for parenchyma-stromal interactions of OSCC

Minor:

Line 43 need reference

Introduction:

Need to state results clearly. Introduction needs to be short, but precise

Major:

Fig. 1

Authors makes assumption of SHH expression as being “strong” expression. However, there is no quantifiable results presented.  Is there a way of quantifying intensity? Protein expression profile should be presented in a quantifiable manner. This is speacially true because these are two different anatomic structures and they both stain for used biomarkers.

Fig. 2: Stroma shown in fig 2a and 2b are not comparable. The focus in 2b is much smaller.

Fig. 3 The magnification for all 3 panels in 3 should be mentioned. Why is merge shown bigger panel than other two panels? What are stains here?

Fig. 4 . See my comments above for 4d and h

Overall, study is done in patient samples and has a lot of potential. However, these results should be corroborated by cell biology and genetics approaches.

Author Response

Takabatake et al in the manuscript titled “The Role of Sonic Hedgehog Signaling in the Tumor  Microenvironment of Oral Squamous Cell Carcinoma“ performed H&E staining, immunohistochemistry (IHC), double IHC, and double-fluorescent IHC on samples of different oral squamous cell carcinoma (OSCC) subtypes to study Sonic hedgehog (SHH), expression of its receptor, Patched-1 (PTCH) in OSCC tumor microenvironment (TME). By using 15 endophytic-type cases (ED type) and 15 exophytic-type cases (EX type) cases they suggest that SHH –autocrine signaling induces growth promotion and cancer invasion. They also suggest that SHH paracrine effects are responsible for parenchyma-stromal interactions of OSCC

Minor:

Line 43 need reference

→We have added to the reference in the Introduction section of the revised manuscript. (p.1, line 43)

Introduction:

Need to state results clearly. Introduction needs to be short, but precise

→We have modified the Results section and we have shorten the introduction section.

Major:

Fig. 1

Authors makes assumption of SHH expression as being “strong” expression. However, there is no quantifiable results presented.  Is there a way of quantifying intensity? Protein expression profile should be presented in a quantifiable manner. This is speacially true because these are two different anatomic structures and they both stain for used biomarkers.

→We have added to the method of quantifying intensity in the Materials and Methods section and have added to the results in Results section of the revised manuscript. (p.4, line 110-120. and p.11, line 368-370)

Fig. 2: Stroma shown in fig 2a and 2b are not comparable. The focus in 2b is much smaller.

→We have modified the figure 2a (right side; EX-type stroma). (p.5, Figure. 2)

Fig. 3 The magnification for all 3 panels in 3 should be mentioned. Why is merge shown bigger panel than other two panels? What are stains here?

→We have added to the scale bar in Fig. 3g .We want to emphasize the merge findings, thus we showed bigger panel than other two panels and we have already shown the CXCR4 and PTCH expression by immunohistochemistry in Fig. 3.

Fig. 4 . See my comments above for 4d and h

→We have added to the scale bar in Fig. 3d, and g. We want to emphasize the merge findings, thus we showed bigger panel than other two panels and we have already shown the CD68 and CD11b expression by immunohistochemistry in Fig. 4.

Overall, study is done in patient samples and has a lot of potential. However, these results should be corroborated by cell biology and genetics approaches.

→We agree that the cell biological experiments and genetics approaches are important line of this study. We have now acknowledged this and suggested it as a topic for further research in the Discussion section of the revised manuscript (p.11, line 331-333)

Reviewer 2 Report

The manuscript examined the expression of Sonic Hedgehog Signaling (SHH) in the tumor microenvironment (TME) of oral Squamous cell carcinoma (OSCC). They should rephrase the conclusions that identified by IHC in OSCC than as noted “autocrine effects of SHH induce cancer invasion and paracrine effects of SHH govern parenchyma-stromal interactions of OSCC. The role of the SHH pathway is to promote growth and invasion”. Otherwise, they need to show more studies to claim autocrine effect, growth and invasion of OSCC. In general, ambitious interpretation and conclusions of the study. Also, a rationale for experiments conducted should be provided appropriately. Although the abbreviations used were defined in “key words”, it would be better to define in subtitles for the readers to follow.

p.2; line 57- “CD11b is a marker of macrophages…” CD11b is expressed on other cell types such as monocytes. Please clarify this as monocyte macrophage lineage specific marker.

p.3; line 98-subtitle: SHH and PTCH expression in OSCC- They should explain the rationale why and how these studies were performed.

p.6, line 163- “SDF-1/CXCL12 plays multiple roles in tumor pathogenesis”—they should give a rationale why non-hedgehog signaling (SHH) molecule like SDF-1 is being studied in Fig.3.

p.8, line 191-TAMs- please define the abbreviation in the subtitle. Also, they should provide a rational for Fig.4 experiments.

p.10, line 231-CAFs- please define the abbreviation in the subtitle.

p.11, line 252-Discussion- “Our study elucidates a novel mechanism of SHH signaling in the TME of OSCC”- please rephrase simply studying immunohistochemistry of these tumors, but not a novel mechanism.

Author Response

Reviewer 2

The manuscript examined the expression of Sonic Hedgehog Signaling (SHH) in the tumor microenvironment (TME) of oral Squamous cell carcinoma (OSCC). They should rephrase the conclusions that identified by IHC in OSCC than as noted “autocrine effects of SHH induce cancer invasion and paracrine effects of SHH govern parenchyma-stromal interactions of OSCC. The role of the SHH pathway is to promote growth and invasion”. Otherwise, they need to show more studies to claim autocrine effect, growth and invasion of OSCC. In general, ambitious interpretation and conclusions of the study. Also, a rationale for experiments conducted should be provided appropriately. Although the abbreviations used were defined in “key words”, it would be better to define in subtitles for the readers to follow.

p.2; line 57- “CD11b is a marker of macrophages…” CD11b is expressed on other cell types such as monocytes. Please clarify this as monocyte macrophage lineage specific marker.

→We have added to the comment in the Introduction section of the revised manuscript. (p.2, line 56)

p.3; line 98-subtitle: SHH and PTCH expression in OSCC- They should explain the rationale why and how these studies were performed.

→We have explained the rationale in the Results section. (p.2, line 93-95)

p.6, line 163- “SDF-1/CXCL12 plays multiple roles in tumor pathogenesis”—they should give a rationale why non-hedgehog signaling (SHH) molecule like SDF-1 is being studied in Fig.3.

→We have mentioned the rationale between SHH and SDF-1in the Introduction section. (p.2, line 62-65. p.2, line 79-80)

p.8, line 191-TAMs- please define the abbreviation in the subtitle. Also, they should provide a rational for Fig.4 experiments.

→We have defined the abbreviation in subtitle, and have mentioned the rationale CD68 and CD11b for this experiments in the Results section. (p.7, line 188-192. p.7, line 201-203)

p.10, line 231-CAFs- please define the abbreviation in the subtitle.

→We have defined the abbreviation in subtitle in the Results section. (p.7, line 214)

p.11, line 252-Discussion- “Our study elucidates a novel mechanism of SHH signaling in the TME of OSCC”- please rephrase simply studying immunohistochemistry of these tumors, but not a novel mechanism.

→We have modified the comment in the Discussion section. (p.7, line228-229)

Round 2

Reviewer 1 Report

I am satisfied with the changes.